# Delayed Surgical Management of Congenital Syndactyly Improves Range of Motion: A Long-Term Follow-Up

**DOI:** 10.3390/jcm14093200

**Published:** 2025-05-05

**Authors:** Aba Lőrincz, Hermann Nudelman, Edina Ilona Kormos, Gergő Józsa

**Affiliations:** 1Department of Thermophysiology, Institute for Translational Medicine, Medical School, University of Pécs, 12 Szigeti Street, H7624 Pécs, Hungary; aba.lorincz@gmail.com (A.L.); nuhwaao.pte@tr.pte.hu (H.N.); 2Institute of Information and Electrical Technology, Faculty of Engineering and Information Technology, University of Pécs, Boszorkány Street 2, H7624 Pécs, Hungary; 3Division of Pediatric Surgery, Traumatology, Urology and Pediatric Otolaryngology, Department of Pediatrics, Medical School, University of Pécs, 7 József Attila Street, H7623 Pécs, Hungary; ebngbg@tr.pte.hu

**Keywords:** pediatric, syndactyly, follow-up, range of motion, ROM, DASH

## Abstract

**Background:** Syndactyly, the congenital fusion of digits, compromises hand function and esthetics. Although surgical separation is the standard treatment, the optimal timing of the intervention remains controversial. **Methods:** We prospectively analyzed 20 pediatric patients (86 operated fingers) undergoing syndactyly repair, comparing early (≤24 months) versus delayed (>24 months) surgery. Outcome measures included range of motion (ROM) at the metacarpophalangeal (MP), proximal interphalangeal (PIP), and distal interphalangeal (DIP) joints; complications (synostosis, nail deformities, finger length disparity, webbing); and patient-reported outcomes assessed by the Disabilities of the Arm, Shoulder, and Hand (DASH) and overall esthetic satisfaction scores. **Results:** The median age at surgery was 31 months (IQR25/75: 24.75–36.5), with a median follow-up of 72 months (IQR25/75: 42.0–86.25). Notably, digits III (28.24%) and IV (29.41%) were predominantly affected. Delayed surgery resulted in significantly improved MP ROM (90.98° ± 8.44° vs. 73.13° ± 22.37°, *p* = 0.004) and DIP ROM (76.28° ± 22.24° vs. 67.19° ± 22.78°, *p* = 0.028), with a non-significant trend toward better PIP ROM (93.00° ± 25.18° vs. 77.37° ± 30.29°, *p* = 0.075). Furthermore, the incidence of synostosis was markedly reduced in the delayed surgery group (6.0% vs. 38.9%, *p* = 0.001). Despite superior joint function associated with delayed intervention, early surgery patients reported higher satisfaction with cosmetic results (3.00 vs. 2.80, *p* = 0.028), while the DASH scores remained comparably low between groups (0.00 vs. 0.24, *p* = 0.141). Finger length disparities and webbing were minimal. **Conclusions:** Our study challenges the conventional advocacy for early syndactyly repair, by demonstrating that delaying surgery beyond 24 months significantly enhances joint mobility and reduces the synostosis rate. However, the higher satisfaction observed as a result of early intervention suggests that surgical timing should be individualized for affected fingers, joints, and severities to balance the functional and cosmetic outcomes. Further studies are needed to define the optimal surgical timing and techniques for pediatric syndactyly.

## 1. Introduction

Syndactyly, the congenital fusion of two or more digits, stands out as a prevalent limb malformation in pediatrics, occurring in roughly 3–109/10.000 births globally [1,2,3,4]. Although toes are involved five times more often than fingers, hand syndactyly imposes far greater functional, psychological, aesthetic, and socioeconomic burdens [4,5,6]. Arising from interrupted digit separation during embryonic development (i.e., webbing), this condition is influenced by a complex interplay of genetic and environmental factors, such as Caucasian heritage, maternal age, smoking, socioeconomic, and nutritional status [2,7].

Limb development begins in week 4 of embryogenesis, when the upper limb bud forms, followed 1–2 days later by the lower limb bud. Anteroposterior (AP) patterning is governed by the zone of polarizing activity (ZPA), while mesenchymal proliferation depends on the apical ectodermal ridge (AER) [3]. By week six, the buds flatten into hand and foot plates, constrictions define regions, and digit separation occurs via AER-induced apoptosis [7]. HOX genes specify the proximal (humerus), intermediate (radius/ulna), and distal (hand) segments; TBX4/5 and PITX1 distinguish lower and upper limb identity, with mutations causing syndactyly, among others [3,4,7].

At the posterior margin, the ZPA produces retinoic acid (RA) to induce sonic hedgehog (SHH) expression and establish the AP axis as it shifts beneath the AER; ectopic RA/SHH causes mirror duplications [7]. Mutations in HOXD13, and less commonly in TTC30B, FBLN1, ZRS, FGF16, and LRP4, disrupt WNT, Fibroblast Growth Factors (FGFs), Bone Morphogenetic Proteins (BMPs), and RA interactions, thereby leading to interdigital cell death in the autopod [7,8,9].

Most cases are isolated, autosomal-dominant anomalies (“non-syndromic”), but syndactyly also appears in syndromic, rare, recessive, and X-linked contexts (e.g., VACTERL association, Apert, Holt–Oram, Jackson–Weiss, Pfeiffer, and Poland syndromes), or alongside other digital malformations, such as acro-, campto-, clino-, or polydactyly [3,8]. Classification hinges on the extent and complexity of fusion: simple (cutaneous) versus complex (osseous); complete, extending the entire digit, or incomplete; halting before the distal phalanx; with complicated (hidden polydactyly) cases exhibiting inter-ray bone formation that influences surgical planning [1,7]. Malik’s modified Temtamy–McKusick system further subdivides nine major non-syndromic phenotypes, with additional subtypes in several groups [10,11].

A thorough preoperative evaluation is essential, incorporating assessments of the patient’s physical, sensory, and motor functions, as well as muscle strength, joint mobility, and cognitive status. Although infant radiographs have limited early diagnostic utility, they may become critical preoperatively, ideally after 10 months of age or later. A two-view hand–forearm X-ray is recommended for suspected bone involvement, supplemented by laboratory tests and, in select cases, angiography and Doppler studies [12,13]. Careful documentation through photography or video footage further informs the operative strategy.

Surgical intervention remains the primary treatment, targeting functional restoration through the precise reconstruction of soft tissues, bones, and nails. Traditionally, an optimal timing of between six months and two years of age was recommended, due to the benefits of early intervention compared to the surgical risks [12,13,14,15,16]. Techniques vary by syndactyly type, frequently necessitating skin grafting, while adjunct physiotherapy, occupational therapy, and orthotics initiated in infancy optimize the outcomes. Simple and incomplete syndactyly are typically managed with digit separation and web space reconstruction, often via Z-plasty and occasionally with full-thickness skin grafts (FTSGs) from the cubital or wrist region, whereas complex cases require osteotomies, joint realignment, and tendon–ligament reconstruction [15,16,17]. Advanced, often staged, procedures are indicated for complete and complicated syndactyly to accommodate growth and minimize scarring, with tailored grafting and flap techniques ensuring effective skin coverage [17,18].

Despite advancements, complications such as joint stiffness, web creep, scar contracture, keloids, and misalignment can impact long-term function and esthetics [12,17,18]. Prognosis varies with severity, depending on the digit shape, length, and motion along with hand, shoulder, and elbow mobility, with complex cases (~10%) requiring revision surgeries [19]. Close follow-up is essential for early detection and intervention in response to complications.

Our study focuses on the long-term surgical functional and esthetic outcomes in terms of different fingers in children with congenital syndactyly, especially based on the timing of the operation and wound closure technique, which are still debated [16,20]. By evaluating demographic and functional outcomes, we aim to provide a thorough overview of the effectiveness and challenges to current surgical approaches in treating this complex condition.

## 2. Materials and Methods

### 2.1. Study Design and Setting

We conducted a cohort study at the Surgical Division, Department of Pediatrics, Medical School, University of Pécs, Pécs, Hungary, focusing on children who underwent surgical treatment for syndactyly. Children who underwent syndactyly repair between January 2013 and January 2024 were retrospectively identified. In February 2024, these patients were recalled for prospective follow-up using goniometry, a physical examination, and a structured questionnaire (Appendix A).

The study was approved by the Institutional Review Board of the Surgical Division, Department of Pediatrics, Medical School, University of Pécs, on 17 January 2024 (PTE/00631-011/2024), and all the procedures followed were in accordance with the ethical standards of the responsible committee on human experimentation and with the Helsinki Declaration of 1975, as revised in 2013.

### 2.2. Participants and Eligibility Criteria

(1) Children (aged ≤ 18 years) with (2) diagnosed congenital syndactyly (3) requiring surgical intervention and (4) participating in the follow-up process at least 2 years after surgery, were included. Patients with (1) missing contact information, (2) who did not attend the follow-up, (3) had revision surgery, or (4) had other hand deformities, were excluded from the study.

### 2.3. Intervention

The creation of a soft, scar-free commissure was the primary surgical aim, using techniques such as Z-shaped incisions, flap rotations, FTSG, Z-plasty, or V–Y-plasty [12]. Early-life operations benefited from improved skin elasticity and reduced subcutaneous fat, often allowing finger separation without grafts. However, in many cases, iatrogenic injury risks necessitated delaying surgery beyond age two [20]. The procedures were performed under general anesthesia, with an Esmarch bandage ensuring bloodlessness, and lasted from 60 min to 2–3 h, depending on the anomaly [15]. Z-shaped, volar- and dorsal triangular interdigital flap deepenings were marked, followed by skin incisions, flap preparation, and finger separation (Figure 1) [14].

During finger separation, the preservation of a nourishing artery proved essential, and any bony or tendon connection distal to the metacarpophalangeal (MP) joint was divided; in severe cases, osteotomy or bone shortening, stabilized with a thin Kirschner (K-) wire, became necessary [12]. Vascular and nerve trunks were exposed and preserved, subcutaneous fat was removed, and skin edges were approximated; if tension existed, an FTSG, harvested from the elbow or wrist, was applied. Management aligned with hand surgery principles, emphasizing bloodlessness, precise incision planning, surgical loupe usage for magnification (2.5–3.2×), stable osteosynthesis, and effective flap rotation or grafting [12,13,15,16].

Postoperatively, for commissure formation, soft gauze strips were placed between the fingers and the hand was loosely wrapped up to the forearm, sometimes with a short splint [21]. Circulation was closely monitored, with adjustments made to the dressing and pain management. Limb immobilization occurred early, with weekly dressing changes. Soft tissue surgeries permitted movement after 2–3 weeks, whereas bone surgeries required six weeks of splinting, until K-wire removal and the initiation of therapy [15]. Patients were typically discharged a few days after surgery [12].

Rehabilitation began preoperatively with physiotherapy and occupational therapy, continued in-hospital, and then at home under parental supervision. Regular check-ups monitored the functional outcomes and determined whether additional surgeries were needed until growth was complete. The patient’s quality of life and hand function were assessed using international Disabilities of the Arm, Shoulder, and Hand (DASH) questionnaires [22].

### 2.4. Evaluated Variables

The surgical outcomes were noted after a physical examination and included range of motion (ROM) evaluations using goniometers in regard to the MP, proximal interphalangeal (PIP), and distal interphalangeal (DIP) joints. Normal ROM flexion intervals for the aforementioned articulations are between 85 and 90 degrees (°), 100 and 115°, and 80 and 90°, respectively. It must be noted that the standard ROM for the pollex (I) is 45–50° for the MP joint and 58–90° for the interphalangeal (IP) joint. Complications, such as finger length differences, the occurrence of synostosis or nail deformity, and webbing post-surgery, were also key endpoints.

Additional outcomes included age at surgery and follow-up, grafting need and donor site, follow-up duration (months from surgery to last follow-up), sex, affected fingers, severity, and anomaly side. Parent satisfaction ratings (1: not satisfied, 2: moderately satisfied, 3: completely satisfied with cosmetic result) and DASH scores were also collected [22]. DASH questionnaires were designed to evaluate the functional impairment and severity of limb musculoskeletal disorders, while reflecting surgical success. The score components included disability/symptoms (30 questions) and optional modules for sports/music or work (4 questions each), addressing additional impairments. At least 27 of 30 questions had to be answered to compute the score, which was summed up, averaged, and transformed into a 0–100 scale, where higher values indicated greater disability. The completion of the optional modules required responses to all four questions.

### 2.5. Data Analysis

Descriptive statistics (means, medians, interquartile ranges (IQRs), standard deviations (SD), ranges, and percentages) were used to summarize the patient demographics, surgical details, and long-term outcomes. Python 3.9.19 (Python Software Foundation, Wilmington, DE, USA), with the pandas, NumPy, scikit-learn, matplotlib, and seaborn libraries, were employed for data handling, analysis, and visualization. For continuous, non-normally distributed variables, the Mann–Whitney U test was used to compare two independent groups, while the Kruskal–Wallis test was applied for comparisons among more than two groups. Student’s t-tests were employed in the case of equal variances. For categorical endpoints, the Chi-square (χ²) test was used to assess differences in proportions between groups. Counts were compared via exposure-adjusted Poisson rate tests. Differences were deemed significant below a *p*-value of 0.05.

## 3. Results

Twenty pediatric syndactyly patients contributed 86 operated fingers for the analyses. Only a slight male predominance was observed when the number of patients was compared (*p* = 0.371). However, when the sex distribution was evaluated by finger count, it revealed a significantly higher syndactyly risk for boys (*p* < 0.001) (Table 1).

At the time of surgery, a median age of 31 months was noted, with a median follow-up interval of six years. Therefore, the mean patient age was 8.28 years at the time of follow-up (Figure 2).

Seven (35% of all) children had bilateral syndactyly, while unilateral cases were chiefly left sided (69.23% of patients), without statistically significant differences between laterality groups. Primarily fingers IV (29.41% of all fingers) and III (28.24%) were affected. Additionally, digits I, II, and V were operated on equally in 14.12% of instances. When using the χ² goodness-of-fit test, fingers IV (*p* = 0.026) and III (*p* = 0.039) seemed significantly more impacted than fingers V, II, and I. Conversely, when the connections between digits were evaluated, their distribution appeared uniform (Table 1).

Regarding severity, fingers were mainly deemed simple, complete (45.52%) and simple, incomplete (29.07%), whereas complex, complete (19.77%), and complicated (4.65%) scenarios were markedly fewer. Simple, complete (*p* < 0.028) fingers were significantly more frequent than all other categories, while simple, incomplete (*p* < 0.001) fingers and complex, complete (*p* = 0.005) fingers were more frequent than complicated fingers.

Our primary endpoints during the control examinations were the functional results for each joint per finger; therefore, the ROM was assessed in all three finger articulations, which are summarized in Figure 3 and Table 2.

Most preserved motions were seen in the PIP joints, with a median of 98°. Furthermore, the MP joint (excluding the pollex) showed a median preserved motion of 89°, while the DIP joint was 80°. The average pollex MP ROM was 45.91°, which is also considered normal. We found physiological and significant differences in the ROM of the IP (average: 72.58°, SD: 25.47°, range: 0–90°) and the PIPs (*p* < 0.022), and between I and other MPs (*p* < 0.019); however, the lower functional boundaries of the pollex are smaller angles.

Despite common beliefs, the later timing of surgery seemed to improve joint functions. When the fingers were grouped by the time of surgery being ≤ or > than 24 months, delayed timing produced wider mean ROMs in the MP (90.98 vs. 73.13°, *p* = 0.004), PIP (93.00 vs. 77.37°, *p* = 0.075), and DIP joints (76.28 vs. 67.19°, *p* = 0.028) when removing I from the analysis (Table 3) (Figure 4).

Building on these kinematic findings, we next examined whether the closure technique and graft-site selection influenced the outcomes. Nineteen patients had their grafting status reported, showing that most children (78.95%) received an FTSG on 73 (84.88% of total) fingers. Chiefly the cubital region (73.97% of transplanted fingers) was selected as the donor site, and the rest of the grafts (26.03%) were harvested from the wrist crease, with one patient receiving grafts from both areas. No graft failure or contracture was noted. As shown in Table 4, direct triangular flap closure was most frequently chosen for the youngest patients, yielding the lowest mean age at surgery (21.69 months) compared to wrist- and elbow-grafted cases (28.11 and 40.28 months, respectively). Therefore, direct closure seemed to correlate with inferior ROM in all the joints, like other early surgeries.

A finger length disparity affected 16.28% of all digits, with a mean of 1.40 mm in every investigated finger, implying a minor degree of post-surgical asymmetry (Figure 3). The average webbing post-surgery was 1.43 mm in all fingers, perceived in 14 children (29.76% of all reported fingers). There was no statistically significant difference between the average webbing (4.76 mm) and finger length difference (8.29 mm) on affected fingers only (*p* = 0.0625), despite webs being 43% smaller on average (Figure 5). The timing of surgery also did not seem to affect digit length or web size (*p* > 0.662) (Table 3). Nevertheless, direct closure showed superior results regarding webbing (0.77 mm vs. <1.42 mm) compared to grafting (*p* < 0.001) (Table 4).

Nail deformities were observed in six patients, involving 11 fingers (Figure 5). Seventeen digits had synostosis, distributed among six patients as follows: a boy with all ten, a boy and girl with two, and three girls with one finger affected. We found a significantly higher occurrence of synostosis than nail deformities (*p* = 0.034) when analyzing all fingers. Surgical timing significantly affected (*p* = 0.002) synostosis likelihood (38.9% in early and 6.0% in delayed interventions) (Table 3). Direct wound closure correlated with an extremely high risk of postoperative synostosis (76.9% vs. <15.8% with grafting, *p* < 0.001), although no nail deformities were found after it was employed. Additionally, a single case of 15° ulnar deviation at the DIP joint of the III digit was observed.

The mean DASH score was 0.32 and the overall esthetic satisfaction score was 2.76. Only two patients had imperfect DASH scores (10% of all the children) and they and another child reported unsatisfactory esthetic results, hinting at generally high degrees of contentment with the surgical results. Despite the inferior functional results, early intervention recipients reported significantly higher (*p* = 0.028) satisfaction rates (early 3.0, delayed 2.8) (Table 3). The DASH ratings were uniform between the wound closure groups; however, imperfect scores were only registered with elbow grafts, which also showed a tendency (*p* = 0.057) towards lower satisfaction levels.

## 4. Discussion

Our findings challenge a long-standing paradigm by demonstrating that delaying syndactyly repair beyond 24 months yielded markedly improved long-term joint mobility compared to earlier interventions. Traditional recommendations, favoring surgery between 6 and 24 months, were primarily based on the benefits of neural plasticity and early digit growth [12,13,14,15,16,17]. Conversely, children who underwent later separation exhibited superior ROM at the MP and DIP joints and a significantly lower incidence of synostosis (Table 3). Early operations seemed to disproportionately compromise motion in all syndactyly configurations, whereas a delayed approach equalized outcomes across different fingers (Figure 4). Postponed interventions may afford the advantage of operating on more mature tissue, with improved neuromuscular control, better-vascularized digits, and greater soft tissue volume, facilitating more precise reconstruction and better compliance during postoperative rehabilitation. Emerging expert consensus advocates that surgical timing should be individualized [20].

Different wound closure techniques also led to significant changes in patient outcomes and were closely related to surgical timing. Direct triangular flap wound closure with sutures was mainly performed in patients younger than two years old, which was associated with the same mobility risks as early interventions (Table 4). Severe synostosis risks during direct closure emphasize the diagnostic difficulties (e.g., immature bones, lower calcification, small structures) when employing interventions early. For grafting, exclusive FTSGs were chosen, mainly from the cubital region, as they offered advantages over split-thickness grafts (STSGs) by shrinking less, growing with the child, being closer to skin texture, and a color match [16,17]. By contrast, a traditional donor site for syndactyly releases has been the groin due to its ample skin supply; however, using groin skin can introduce aesthetic drawbacks [23]. Groin-derived grafts are often thicker, hair-bearing, or hyperpigmented, leading to a noticeable color mismatch on the finger. While recent meta-analyses affirmed FTSG over STSG functional superiority, they questioned its esthetic efficacy and highlighted its higher contracture and rejection rates after hand burn reconstruction and radial forearm flap closures [24]. Alternative techniques, such as dorsal hexagonal flaps, could have reduced grafting requirements [13], and double, opposing rectangular fingertip skin flaps may yield superior cosmetic and functional outcomes, especially regarding nail deformities [15].

Notably, we did not observe statistically substantial finger growth abnormalities or joint angulation in the delayed intervention group, countering concerns that waiting invariably leads to asymmetric development of the fused digits. De Smet et al. found that the majority of web creeps were associated with early interventions; however, we did not observe a substantial difference in the web creep extent [25]. A 2024 analysis of 52 patients aged 2–12 years demonstrated that postponed surgery did not compromise functional or aesthetic outcomes, with improved web spacing and minimal contracture observed, even in older cohorts [26]. Minor finger length disparities and webbing were observed post-surgery, implying that while surgical techniques are generally effective, there remains room to refine surgical precision to minimize these asymmetries [15].

Consistent, generally high contentment levels emphasize the effectiveness of the current surgical techniques in meeting patient and parental expectations. Ferrari and Werker found no difference in patient contentment when investigating grafted versus directly closed syndactyly, similar to our results [18]. Conversely, in this study, early intervention patients reported slightly higher, yet statistically significant, cosmetic satisfaction, suggesting that while delayed surgery enhances functional outcomes, early repair may yield better esthetic results. Significantly higher DASH scores, slightly more nail deformities, and larger finger length disparities were observed in the case of postponed interventions, which seemed to more greatly influence patient satisfaction than functional advantages. Although our study supports delayed repair for improved joint mobility and reduced synostosis, it is crucial to acknowledge that neurodevelopmental benefits from early surgery, such as early adaptation to hand function, may still hold value for some patients, as underscored by the increased average DASH score regarding late interventions. Therefore, an optimal maximum age for postponed operations should also be established.

While this study provides valuable insights, it is not without limitations. Its relatively small sample size and the single-center nature of the analysis limit the generalizability of our findings. Although our study suggests that delayed surgery may offer better functional outcomes, this observation warrants caution. Finger- and joint-specific severities must be directly compared regarding the surgical timing and type of intervention, to further substantiate our claims. Due to the relative rarity and diverse, unique manifestations of this condition, and because the operations took place before this study, the groups were created after collecting the results of the follow-up examinations; therefore, bias in the selection of the reported results may be present. Predominant FTSG use highlighted a preference for specific donor sites, which resulted in intervention selection bias; however, direct closure is only feasible in tension-free environments. Follow-up periods varied significantly among the patients, which could introduce bias into the assessment of the long-term outcomes. Moreover, the commonality of male, left sided, simple, complete, III–IV syndactyly in various populations, limited the drawable conclusions from rarer syndactyly types and suggested potentially unexplored underlying genetic influences [1,3,12]. Regrettably, advanced imaging, blinded, objective scar, and internationally accepted psychosocial evaluation tools were not employed, which could have provided a more holistic understanding of the surgical outcomes. Future studies should aim to include larger, more diverse, randomized, prospective, multicenter cohorts, comparing the available surgical techniques, with standardized and blinded follow-up protocols to validate these preliminary findings.

## 5. Conclusions

Delaying syndactyly repair beyond 24 months with grafted rather than direct flap closure significantly enhanced functional outcomes of the children, evidenced by greater MP and DIP joint ROMs and reduced synostosis rate. Although postponed reconstruction improved patient function, early intervention was associated with marginally higher esthetic satisfaction, indicating a trade-off between mobility and cosmetic outcomes. A lower likelihood of nail deformities, smaller digit length discrepancies, and significantly minor web creeps were noted employing early, direct closures; however, web sizes were slightly smaller after delayed surgeries when compared to early interventions. While our findings challenge conventional early intervention paradigms and advocate for an individualized approach, the study limitations warrant further prospective research. Future work should focus on refining surgical techniques, improving patient counseling, and optimizing care protocols for congenital syndactyly.

## Figures and Tables

**Figure 1 jcm-14-03200-f001:**
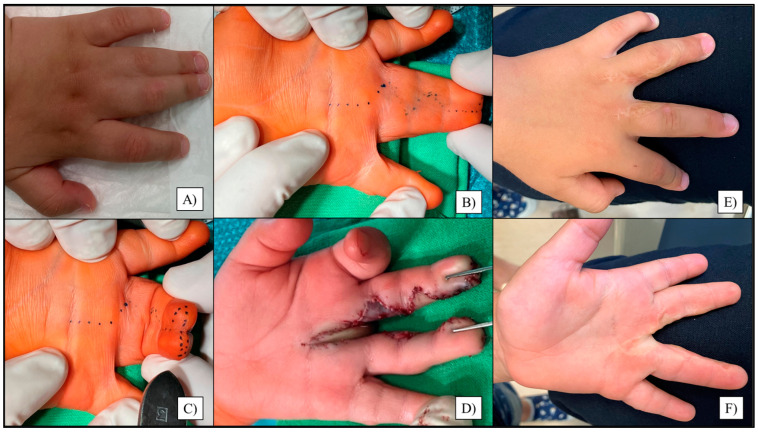
Digital photographs of a left congenital III–IV finger syndactyly of a 2-year-old boy (**A**). Z-shaped incisions were marked preoperatively shown in extended (**B**) and flexed positions (**C**). Postoperative image after finger separation using triangular interdigital flaps (**D**). On the three-year follow-up, the patient showed excellent functional and cosmetic results, with only minor scarring from the dorsal (**E**) and palmar aspects (**F**).

**Figure 2 jcm-14-03200-f002:**
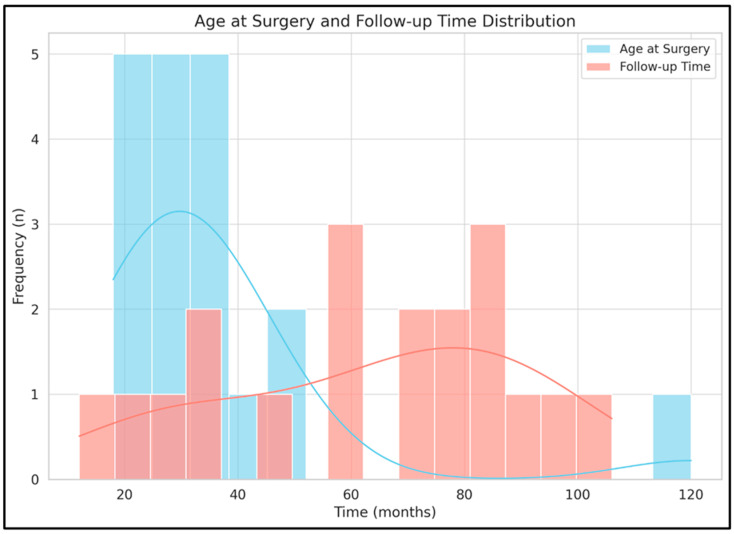
Patient age at the time of surgery and follow-up duration distribution. Bars on the histogram depict actual values, while trendlines represent the expected values.

**Figure 3 jcm-14-03200-f003:**
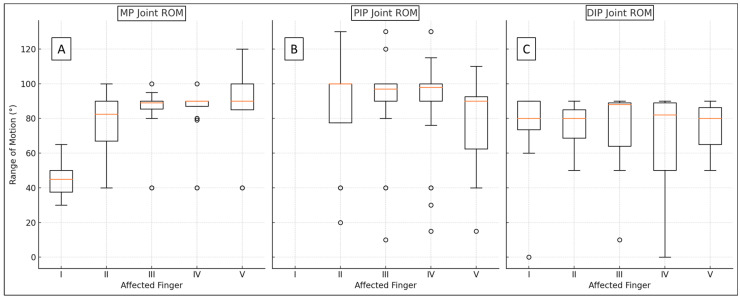
Range of motion (ROM) boxplots divided by joint (MP (**A**), PIP (**B**), DIP (**C**)) and finger. Red lines illustrate the medians, the tops and bottoms of the box depict the IQR75s and -25s, whiskers represent the ranges, and the outliers are expressed by dots.

**Figure 4 jcm-14-03200-f004:**
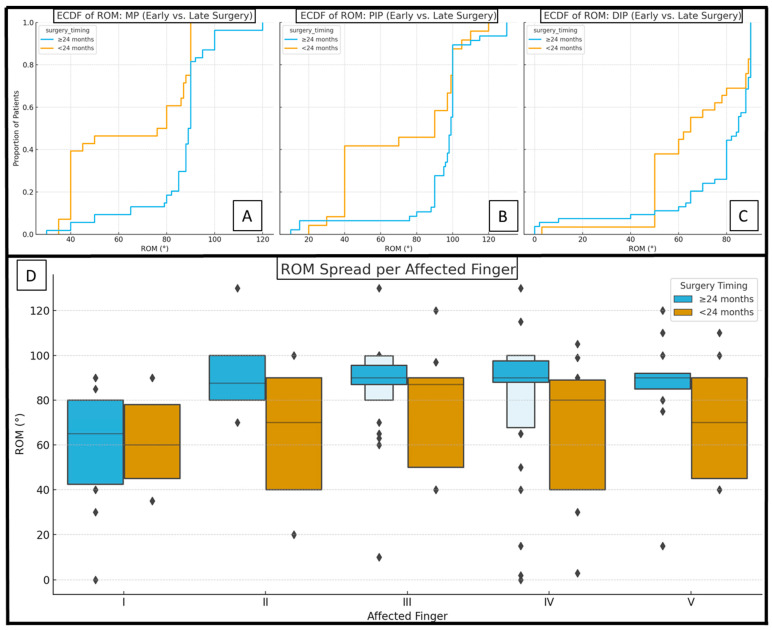
Empirical cumulative distribution functions (ECDFs) of ROMs organized by early and late surgical timing categories and joints (MP (**A**), PIP (**B**), and DIP (**C**)). Box plots (**D**) represent ROM distributions separated by operation timing and affected finger. Each successive level outward contains half of the remaining data.

**Figure 5 jcm-14-03200-f005:**
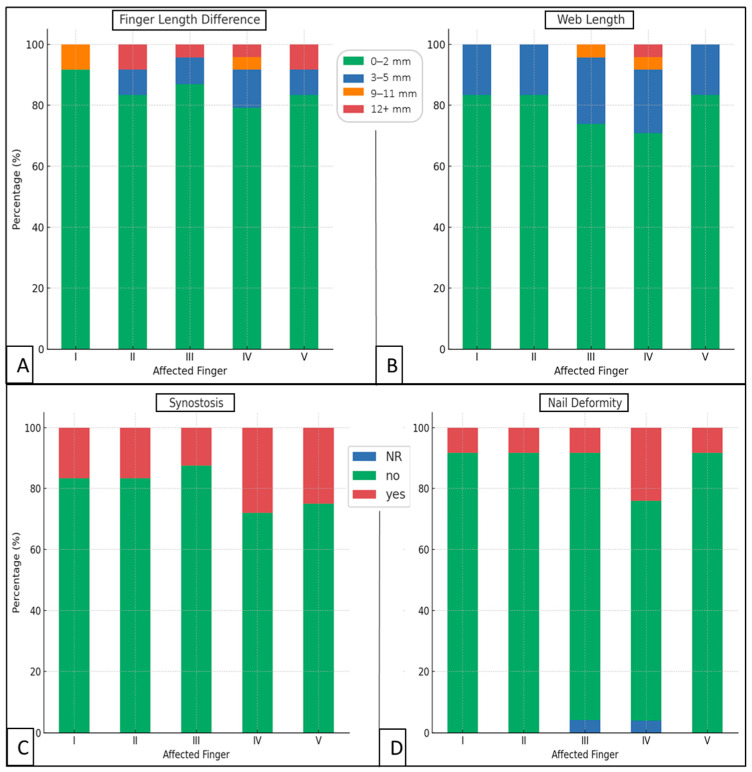
Differences in finger length (**A**), web size (**B**), synostosis (**C**), and nail deformity (**D**) shown as stacked bar plots, demonstrating the relative distributions. NR reflects non-reported status.

**Table 1 jcm-14-03200-t001:** Discrete outcome summary for the whole cohort.

Outcome	Variable	Unit (Total)	Count	Percentage	Notes
sex distribution	boy	patients (*n* = 20)	12	60.00	*p* = 0.371, non-significant
	girl		8	40.00	
	boy	fingers (*n* = 86)	60	69.76	
	girl		26	30.24	*p* < 0.001
laterality	bilateral	patients (*n* = 20)	7	35.00	bilateral vs. unilateral: *p* = 0.263
	unilateral		13	65.00	
	left sided		9	69.23	unilateral cases: *p* = 0.267
	right sided		4	30.77	
digit involvement	IV	fingers (*n* = 85)	25	29.41	*p* = 0.026 vs. I, II, V
	III		24	28.23	*p* = 0.039 vs. I, II, V
	I		12	14.12	
	II		12	14.12	
	V		12	14.12	
interdigital connections	III–IV	connections (*n* = 50)	14	28.00	*p* = 0.940 between groups
	IV–V		13	26.00	
	I–II		12	24.00	
	II–III		11	22.00	
severity	simple, complete	fingers (*n* = 86)	40	45.52	*p* < 0.028 vs. others
	simple, incomplete		25	29.07	*p* < 0.001 vs. complicated
	complex, complete		17	19.77	*p* = 0.005 vs. complicated
	complicated		4	4.65	
wound closure	FTSG	fingers (*n* = 85)	72	84.71	*p* < 0.001 vs. direct
	direct closure		13	15.29	four patients, 21.05% of all patients
FTSG donor site	cubital region	fingers (*n* = 72; 84.88% of all)	53	73.97	one patient had both sites
	wrist crease		19	26.03	utilized, in six children
nail deformities	affected	fingers (*n* = 83)	11	13.25	
synostosis	affected	fingers (*n* = 85)	17	20.00	*p* = 0.034 vs. nail deformities

**Table 2 jcm-14-03200-t002:** Evaluation of continuous outcomes per finger, in brief.

Continuous Outcomes (Unit)	Count	Mean	SD	Min	Max	Median	IQR	IQR25	IQR75
age at surgery (months)	86	33.87	24.63	18	120	31	11.75	24.75	36.5
follow-up time (months)	86	67.82	22.82	12	106	72	44.25	42	86.25
age at follow-up (years)	86	8.28	2.82	3.67	16	8.25	3.79	6.13	9.92
II–V MP (°)	71	83.44	18.08	40	120	89	5	85	90
pollex MP (°)	11	45.91	11.58	30	65	45	13	38	50
PIP (°)	71	86.39	28.33	10	130	98	10	90	100
DIP (°)	83	72.34	22.79	0	90	80	27	63	89
web (mm)	83	1.43	2.74	0	15	0	2	0	2
web * (mm)	25	4.76	3.02	2	15	5	2	3	5
finger length difference (mm)	83	1.40	3.68	0	15	0	0	0	0
finger length disparity * (mm)	14	8.29	4.87	2	15	5	8.75	5	13.75
DASH (x/100)	75	0.32	0.90	0	3.3	0	0	0	0
satisfaction (1–3)	76	2.76	0.56	1	3	3	0	3	3

* Affected fingers only.

**Table 3 jcm-14-03200-t003:** Effect of surgical timing on different outcomes.

**Continuous Outcomes** **(Unit)**	**Early** **(Average ± SD (*n*))**	**Delayed** **(Average ± SD (*n*))**	***p*-Value**
age at surgery (months)	20.31 ± 2.57 (36)	46.08 ± 28.95 (50)	<0.001 *
follow-up time (months)	64.39 ± 18.26 (36)	70.90 ± 26.11 (50)	0.059
II–V MP (°)	73.13 ± 22.37 (30)	90.98 ± 8.44 (41)	0.004 *
pollex MP (°)	41.00 ± 6.52 (5)	50.00 ± 13.78 (6)	0.215
PIP (°)	77.37 ± 30.29 (30)	93.00 ± 25.18 (41)	0.075
DIP (°)	67.19 ± 22.78 (36)	76.28 ± 22.24 (47)	0.028 *
web (mm)	1.56 ± 3.07 (36)	1.34 ± 2.48 (47)	0.937
finger length difference (mm)	1.25 ± 3.02 (36)	1.51 ± 4.14 (47)	0.662
DASH (x/100)	0.00 ± 0.00 (26)	0.24 ± 0.81 (49)	0.141
satisfaction (1–3)	3.00 ± 0.00 (27)	2.80 ± 0.50 (49)	0.028 *
**Discrete Outcomes**	**Early (yes/*n* (%))**	**Delayed (yes/*n* (%))**	***p*-Value**
finger	36/86 (41.86)	50/86 (58.14)	0.404
female	14/36 (38.88)	12/50 (24.00)	0.232
grafted	18/29 (62.07)	48/50 (96.00)	<0.001 *
synostosis	14/36 (38.88)	3/50 (6.00)	0.002 *
nail deformity	3/36 (8.33)	8/50 (16.00)	0.229

* Statistically significant difference.

**Table 4 jcm-14-03200-t004:** Effect of wound closure choice on each variable.

**Continuous** **Outcome (Unit)**	**Elbow** **(Average ± SD (*n*))**	**Wrist** **(Average ± SD (*n*))**	**Direct Closure** **(Average ± SD (*n*))**	***p*-Value**
age at surgery (months)	40.28 ± 30.55 (53)	28.11 ± 7.19 (19)	21.69 ± 9.01 (13)	0.027 *
follow-up time (month)	64.23 ± 20.54 (53)	89.42 ± 12.04 (19)	45.15 ± 9.86 (13)	<0.001 *
II–V MP (°)	89.27 ± 11.18 (45)	88.12 ± 4.79 (16)	49.70 ± 20.45 (10)	<0.001 *
pollex MP (°)	42.50 ± 7.58 (6)	60.00 ± 8.66 (3)	35.00 ± 0.00 (2)	0.011 *
PIP (°)	93.27 ± 22.91 (45)	88.75 ± 29.86 (16)	51.70 ± 24.67 (10)	<0.001 *
DIP (°)	74.80 ± 24.00 (51)	73.74 ± 21.47 (19)	60.62 ± 16.78 (13)	0.128
web (mm)	1.61 ± 2.66 (51)	1.42 ± 3.42 (19)	0.77 ± 1.88 (13)	<0.001 *
finger length difference (mm)	1.20 ± 3.53 (51)	1.84 ± 4.78 (19)	1.54 ± 2.40 (13)	0.802
DASH (x/100)	0.22 ± 0.78 (53)	0.00 ± 0.00 (19)	0.00 ± 0.00 (3)	0.434
satisfaction (1–3)	2.77 ± 0.52 (44)	3.00 ± 0.00 (19)	3.00 ± 0.00 (13)	0.057
**Discrete** **Outcomes**	**Elbow** **(yes/*n* (%))**	**Wrist** **(yes/*n* (%))**	**Direct Closure** **(yes/*n* (%))**	***p*-Value**
fingers	53/85 (62.35%)	19/85 (22.35%)	13/85 (15.30%)	<0.001 *
female	20/53 (37.74%)	2/19 (10.53%)	3/13 (23.08%)	0.071
synostosis	4/53 (7.55%)	3/19 (15.79%)	10/13 (76.92%)	<0.001 *
nail deformity	8/53 (15.09%)	3/19 (15.79%)	0/13 (0.00%)	0.319

* Statistically significant difference.

## Data Availability

The data underlying this study include crucial sensitive information (birth and follow-up dates) and are not publicly available due to current institutional and governmental restrictions. For further inquiries, please contact the corresponding author.

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
