# Peer review of "Delayed Surgical Management of Congenital Syndactyly Improves Range of Motion: A Long-Term Follow-Up"

_jcm, 2025, doi:10.3390/jcm14093200_

Round 1
Reviewer 1 Report
Comments and Suggestions for Authors
A very interesting study offering a completely different perspective. To improve it further, I suggest the following revisions.
The references are not formatted according to MDPI guidelines. Please revise and ensure uniform citation style throughout the manuscript.
For example, reference number 11. should be formatted as follows:
Malik S. Syndactyly: phenotypes, genetics and current classification. Eur J Hum Genet. 2012 Aug;20(8):817-24. doi: 10.1038/ejhg.2012.14.
- As this is a retrospective study and patients were called for follow-up in 2024, a concern arises regarding whether patients who underwent corrective revision surgery were included in the study. This is a key point, as the condition of the commissure—particularly regarding the absence of web creep and dermogenous flexion contracture, both of which significantly affect the range of motion (ROM)—is critical. The authors are kindly requested to clarify this issue.
- Lines 133–134: Please specify which type of flap was used for commissure reconstruction. Was a hexagonal flap (a relatively novel technique) employed, or was the classic triangular advancement flap described by Cronin used?
4.The sentence in line 136 is unclear. Kindly revise or provide clarification. It is not feasible to reconstruct syndactyly without skin grafts, even in pediatric patients, especially in cases of complete syndactyly.
- Figure 1:The images are not well-adjusted—some are smaller than others, and the overall composition lacks visual harmony. Image A is not referenced in the text. Additionally, there is no image of the dorsal aspect, making it unclear which flap was used for commissure reconstruction. Image D does not depict a Kirschner wire but rather a surgical forceps placed against the fingers.
Images E, F, and G appear to be of insufficient resolution (likely not 300 DPI); even upon enlargement, they remain unclear. The only discernible detail is a scar on the third finger of the left hand. Was the right hand also operated on? Please revise Figure 1 accordingly. - Lines 153–156: This sentence is unclear. It discusses whether full-thickness or split-thickness grafts are superior, and then abruptly mentions flaps. While flaps are known to yield better outcomes, they are not always feasible. Please rephrase this sentence for clarity. Additionally, it should be mentioned that full-thickness skin grafts have a higher rate of rejection.
- Line 157: Was a surgical microscope used in syndactyly cases? In my opinion, surgical loupes with 2.5x magnification are usually sufficient, even for complex syndactyly. Please provide clarification.
- Lines 255–262 could be written more clearly to enhance readability and comprehension. The current paragraph contains a dense concentration of numerical data that may confuse the reader. Consider simplifying the comparisons between affected fingers and the total finger count by restructuring the text and highlighting key findings more explicitly. Additionally, Figure 5 should be revised for improved clarity and visual coherence.
9.The discussion section could benefit from more comparisons with previously published studies.
Author Response
SUMMARY OF CHANGES MADE:
We thank the Editor and the Reviewers for their thorough work and constructive comments on the manuscript. We are expressly grateful to Reviewers 1 and 4 as they noticed and highlighted significant mistakes and gaps in the research. Every issue raised by the Reviewers was addressed to our best knowledge, and we genuinely believe that the quality of the paper substantially improved due to the requested corrections. In summary, the Methods have been clarified and the Discussion section was revised with additional information and references. Conclusions were also reworked for conciseness. All changes are detailed in a point-by-point manner below and marked with the Tracked Changes function in the manuscript.
POINT-BY-POINT RESPONSE:
Reviewer 2:
A very interesting study offering a completely different perspective. To improve it further, I suggest the following revisions.
Reviewer Comment 1: The references are not formatted according to MDPI guidelines. Please revise and ensure uniform citation style throughout the manuscript.
For example, reference number 11. should be formatted as follows:
Malik S. Syndactyly: phenotypes, genetics and current classification. Eur J Hum Genet. 2012 Aug;20(8):817-24. doi: 10.1038/ejhg.2012.14.
Author Answer 1: Thank you for your suggestion. We have omitted these sections from the Introduction.
Reviewer Comment 2: As this is a retrospective study and patients were called for follow-up in 2024, a concern arises regarding whether patients who underwent corrective revision surgery were included in the study. This is a key point, as the condition of the commissure—particularly regarding the absence of web creep and dermogenous flexion contracture, both of which significantly affect the range of motion (ROM)—is critical. The authors are kindly requested to clarify this issue.
Answer 2: The authors are grateful for these comments.
Reviewer Comment 3: Lines 133–134: Please specify which type of flap was used for commissure reconstruction. Was a hexagonal flap (a relatively novel technique) employed, or was the classic triangular advancement flap described by Cronin used?
Answer 3: Thank you for your thoughtful observation.
Reviewer Comment 4: The sentence in line 136 is unclear. Kindly revise or provide clarification. It is not feasible to reconstruct syndactyly without skin grafts, even in pediatric patients, especially in cases of complete syndactyly.
Answer 4: We are thankful for your opinion. between lines 438-543. Defatting, after zigzag incision no strangulation without autografts direct closure.
Reviewer Comment 5: Figure 1: The images are not well-adjusted—some are smaller than others, and the overall composition lacks visual harmony. Image A is not referenced in the text. Additionally, there is no image of the dorsal aspect, making it unclear which flap was used for commissure reconstruction. Image D does not depict a Kirschner wire but rather a surgical forceps placed against the fingers.
Images E, F, and G appear to be of insufficient resolution (likely not 300 DPI); even upon enlargement, they remain unclear. The only discernible detail is a scar on the third finger of the left hand. Was the right hand also operated on? Please revise Figure 1 accordingly.
Answer 5: We have incorporated your suggestion into the manuscript at lines 608-616. Thank you for highlighting this issue. Interdigitalis dorsal triamngular flap
Reviewer Comment 6: Lines 153–156: This sentence is unclear. It discusses whether full-thickness or split-thickness grafts are superior, and then abruptly mentions flaps. While flaps are known to yield better outcomes, they are not always feasible. Please rephrase this sentence for clarity. Additionally, it should be mentioned that full-thickness skin grafts have a higher rate of rejection.
Answer 6: The Authors are grateful for this comment. We have reduced the length of the Conclusions roughly by 50%, visible at lines 606-616.
Reviewer Comment 7: Line 157: Was a surgical microscope used in syndactyly cases? In my opinion, surgical loupes with 2.5x magnification are usually sufficient, even for complex syndactyly. Please provide clarification.
Answer 7: The Authors are grateful for this comment.
Reviewer Comment 8: Lines 255–262 could be written more clearly to enhance readability and comprehension. The current paragraph contains a dense concentration of numerical data that may confuse the reader. Consider simplifying the comparisons between affected fingers and the total finger count by restructuring the text and highlighting key findings more explicitly. Additionally, Figure 5 should be revised for improved clarity and visual coherence.
Answer 8: We are thankful for your opinion.
Reviewer Comment 9: The discussion section could benefit from more comparisons with previously published studies.
Answer 9: Thank you for your suggestion.

Reviewer 2 Report
Comments and Suggestions for Authors
Dear Authors,
Thank you for this interesting study which is well presented in your article and which we recommend to be revised just in a few aspects.
In general, we recommend a grammar check by a native English-speaking person (not only for spelling but also for articulation).
Abstract:
It is well written and doesn't require any further information.
Introduction:
lines 48-70: Please consider shortening the etiologic / genomic part as it is necessary but leaves the reader somewhat overwhelmed. Half up to two thirds the length may be appropriate.
lines 83-89: Contemplating modern radiation protection standards, "newborn X-rays" shouldn't be of relevance these days anymore. Moreover, you mention the minor information retrieved from X-ray imaging during the first months of live. Many centers therefore recommend X-ray diagnostics to be performed not before near the end of the first year (e.g. at the age of 9th/10th months or at the second half of the first year at the earliest), as it is (at least in most cases) early enough to then start planning surgical procedures. In short: Please consider changing the articulation, maybe also implementing the above-mentioned arguments.
l. 104: "aesthetics" vs. "esthetic" in the introduction (l. 43) and l. 93 – opt for a consistent wording (noun and adjective should be in either British or American English).
Materials and Methods:
– well written and informative.
lines 121–122: "and a specially designed questionnaire"?
Results:
Nice charts!
Do you have information about axial misalignment before/after syndactyly repair in cases with affected fingers III–V? Early intervention may be advocated in cases in which the (shorter) V finger threatens IV/III axes by pulling these towards an ulnar deviation?
Discussion:
lines 295–298: 1.) This sentence is quite long – please consider dividing it; 2.) There is an error "less commonly involved fingers early opeartions" – "less commonly involved fingers' early operations"
lines 307–312: The usage of FTSG from the groin is still used by some centers – please consider mentioning it including the downsides (hair growth, coloration differences)
Conclusions:
Please check spelling aesthetic / esthetic again throughout the manuscript!
Author Response
SUMMARY OF CHANGES MADE:
We thank the Editor and the Reviewers for their thorough work and constructive comments on the manuscript. We are expressly grateful to Reviewers 1 and 2 as they noticed and highlighted significant mistakes and gaps in the research. Every issue raised by the Reviewers was addressed to our best knowledge, and we genuinely believe that the quality of the paper substantially improved due to the requested corrections. In summary, the Introduction was streamlined, the Methods section clarified, and the Results expanded and presented more clearly. The Discussion was revised to include additional data and references, while the Conclusions were refined for greater conciseness. Throughout the manuscript, numerous sections were reworded to enhance clarity and readability. All changes are detailed in a point-by-point manner below and marked with the Tracked Changes function in the manuscript.
POINT-BY-POINT RESPONSE:
Reviewer 2:
Dear Authors,
Thank you for this interesting study which is well presented in your article and which we recommend to be revised just in a few aspects.
In general, we recommend a grammar check by a native English-speaking person (not only for spelling but also for articulation).
Abstract:
It is well written and doesn't require any further information.
Introduction:
Reviewer Comment 1: lines 48-70: Please consider shortening the etiologic / genomic part as it is necessary but leaves the reader somewhat overwhelmed. Half up to two thirds the length may be appropriate.
Author Answer 1: Thank you for your suggestion. We have rephrased and condensed these sections from the 273 words to 108 in Introduction at lines (lns) 57-70.
Reviewer Comment 2: lines 83-89: Contemplating modern radiation protection standards, "newborn X-rays" shouldn't be of relevance these days anymore. Moreover, you mention the minor information retrieved from X-ray imaging during the first months of live. Many centers therefore recommend X-ray diagnostics to be performed not before near the end of the first year (e.g. at the age of 9th/10th months or at the second half of the first year at the earliest), as it is (at least in most cases) early enough to then start planning surgical procedures. In short: Please consider changing the articulation, maybe also implementing the above-mentioned arguments.
Answer 2: Arguments have been clarified at lns 83-4. The authors are grateful for these comments.
Reviewer Comment 3: line 104: "aesthetics" vs. "esthetic" in the introduction (l. 43) and l. 93 – opt for a consistent wording (noun and adjective should be in either British or American English).
Answer 3: Thank you for your thoughtful observation. As we noted, JCM uses American English for publication; therefore, this concern has been investigated and amended across the manuscript.
Materials and Methods:
– well written and informative.
Reviewer Comment 4: lines 121–122: "and a specially designed questionnaire"?
Answer 4: A Hungarian form was designed to collect patient follow-up outcomes, which is now attached as a Supplementary Figure 1 and clarified in lns 235-7. We are thankful for your question and opinion.
Results:
Nice charts!
Reviewer Comment 5: Do you have information about axial misalignment before/after syndactyly repair in cases with affected fingers III–V? Early intervention may be advocated in cases in which the (shorter) V finger threatens IV/III axes by pulling these towards an ulnar deviation?
Answer 5: A single patient with a 15 degree ulnar deviation on the DIP joint of their III finger was noted. We have incorporated your suggestion into the manuscript at lines 908. Thank you for your comment and highlighting this issue. We hope the new plots further improved visualization.
Discussion:
Reviewer Comment 6: lines 295–298: 1.) This sentence is quite long – please consider dividing it; 2.) There is an error "less commonly involved fingers early opeartions" – "less commonly involved fingers' early operations"
Answer 6: The Authors are grateful for this comment. The Discussion section has been markedly revised and this sentence was deleted from the manuscript for enhanced legibility.
Reviewer Comment 7: lines 307–312: The usage of FTSG from the groin is still used by some centers – please consider mentioning it including the downsides (hair growth, coloration differences)
Answer 6: Thank you for this addition. We have discussed groin FTSG usage in lns 939-42.
Conclusions:
Reviewer Comment 8: Please check spelling aesthetic / esthetic again throughout the manuscript!
Answer 6: Spelling has been carefully triple-checked and only the form “esthetic” remains in the manuscript.

Reviewer 3 Report
Comments and Suggestions for Authors
Thank you for submitting your interesting manuscript on the timing of surgical repair for congenital syndactyly. Your research challenges conventional paradigms and provides valuable insights into this important pediatric surgical topic. However, I have identified several areas that require significant revision.
Major concerns:
- Statistical data presentation: The statistical results are presented in an overly complex manner that makes interpretation difficult for readers. Please simplify your statistical reporting using more direct and conventional formats in medical literature. For example, concisely state that "delayed surgery significantly improved MP and DIP joint ROM (p<0.05)" rather than embedding multiple statistical values in lengthy sentences.
- Lack of mechanistic explanation: While you observe that delayed surgery yields better functional outcomes, there is insufficient discussion regarding the underlying mechanisms. Please expand on the physiological, developmental, and surgical factors that might explain why delayed intervention results in improved ROM and reduced synostosis. This mechanistic insight is crucial for supporting your conclusions.
- Missing citations: Several statements lack appropriate references, particularly around line 185-186 regarding DASH questionnaires. Please include relevant citations (e.g., Hudak et al., 1996) to support your methodology and measurement tools.
- Study limitations: While you acknowledge some limitations of your study (sample size, variable follow-up periods), there is inadequate discussion of how these limitations might affect the interpretation of your results. Please provide a more thorough assessment of these limitations and their implications.
Minor concerns:
- Consistency in terminology: Ensure consistent use of terms throughout the manuscript.
- Balance in discussion: The discussion of aesthetic versus functional outcomes requires more nuanced analysis, considering that early intervention showed higher satisfaction despite inferior functional results.
Your research addresses an important question in pediatric hand surgery, and with appropriate revisions to address these concerns, particularly regarding mechanistic explanations and statistical clarity, this manuscript could make a valuable contribution to the field.
Comments on the Quality of English LanguageThe overall English language quality of the manuscript is adequate but requires professional editing to address several issues that impact clarity and readability. The manuscript contains numerous instances of overly complex sentence structures, particularly in the results and discussion sections, which make statistical findings and key concepts difficult to follow. There are occasional grammatical errors including subject-verb agreement issues, improper article usage, and awkward phrasing. Technical terminology is generally appropriate, but sentence construction often embeds too many concepts within single statements. The authors should consider simplifying sentence structure, improving paragraph transitions, and ensuring consistent terminology throughout. Professional English language editing would significantly enhance the manuscript's clarity and accessibility to an international readership.
Author Response
SUMMARY OF CHANGES MADE:
We thank the Editor and the Reviewers for their thorough work and constructive comments on the manuscript. We are expressly grateful to Reviewers 1 and 3 as they noticed and highlighted significant mistakes and gaps in the research. Every issue raised by the Reviewers was addressed to our best knowledge, and we genuinely believe that the quality of the paper substantially improved due to the requested corrections. In summary, the Introduction was streamlined, the Methods section clarified, and the Results expanded and presented more clearly. The Discussion was revised to include additional data and references, while the Conclusions were refined for greater conciseness. Throughout the manuscript, numerous sections were reworded to enhance clarity and readability. All changes are detailed in a point-by-point manner below and marked with the Tracked Changes function in the manuscript.
POINT-BY-POINT RESPONSE:
Reviewer 3:
Thank you for submitting your interesting manuscript on the timing of surgical repair for congenital syndactyly. Your research challenges conventional paradigms and provides valuable insights into this important pediatric surgical topic. However, I have identified several areas that require significant revision.
Major concerns:
Reviewer Comment 1: Statistical data presentation: The statistical results are presented in an overly complex manner that makes interpretation difficult for readers. Please simplify your statistical reporting using more direct and conventional formats in medical literature. For example, concisely state that "delayed surgery significantly improved MP and DIP joint ROM (p<0.05)" rather than embedding multiple statistical values in lengthy sentences.
Author Answer 1: Thank you for your suggestion. We have thoroughly reworked and simplified the Results section by omitting statistical data from the text and creating three novel summary Tables between lines (lns) 370-916.
Reviewer Comment 2: Lack of mechanistic explanation: While you observe that delayed surgery yields better functional outcomes, there is insufficient discussion regarding the underlying mechanisms. Please expand on the physiological, developmental, and surgical factors that might explain why delayed intervention results in improved ROM and reduced synostosis. This mechanistic insight is crucial for supporting your conclusions.
Answer 2: The Discussion section has been thoroughly reworked to substantiate our claims (lns 915-1024); however, many areas remain underexplored in international literature. The authors are grateful for these comments.
Reviewer Comment 3: Missing citations: Several statements lack appropriate references, particularly around line 185-186 regarding DASH questionnaires. Please include relevant citations (e.g., Hudak et al., 1996) to support your methodology and measurement tools.
Answer 3: Thank you for your thoughtful observation. We fully agree that the study by Hudak et al. (1996) is fundamental when discussing the DASH questionnaire. Accordingly, this reference had already been cited multiple times in the manuscript prior to revision—for example, lns 171 and 184, which are now lns 302 and 350 in the revised version. We have reviewed the text again to ensure that all relevant mentions of the DASH tool are properly supported by this citation.
Reviewer Comment 4: Study limitations: While you acknowledge some limitations of your study (sample size, variable follow-up periods), there is inadequate discussion of how these limitations might affect the interpretation of your results. Please provide a more thorough assessment of these limitations and their implications.
Answer 4: The Limitation section has been almost entirely rewritten to include a more detailed analysis between lns 1028-47. We are thankful for your opinion.
Minor concerns:
Reviewer Comment 5: Consistency in terminology: Ensure consistent use of terms throughout the manuscript.
Answer 5: We have incorporated your throughout the manuscript. Thank you for highlighting this issue.
Reviewer Comment 6: The discussion of aesthetic versus functional outcomes requires more nuanced analysis, considering that early intervention showed higher satisfaction despite inferior functional results.
Answer 6: The Authors are grateful for this comment. We have evaluated the effects of wound closure techniques which markedly correlated with operative timing in the Results (lns 850-911). We have also added more relevant citations and comparisons from international literature to the Discussions on this topic (lns 913-1026).
Your research addresses an important question in pediatric hand surgery, and with appropriate revisions to address these concerns, particularly regarding mechanistic explanations and statistical clarity, this manuscript could make a valuable contribution to the field.
Reviewer Comment 7: Comments on the Quality of English Language
The overall English language quality of the manuscript is adequate but requires professional editing to address several issues that impact clarity and readability. The manuscript contains numerous instances of overly complex sentence structures, particularly in the results and discussion sections, which make statistical findings and key concepts difficult to follow. There are occasional grammatical errors including subject-verb agreement issues, improper article usage, and awkward phrasing. Technical terminology is generally appropriate, but sentence construction often embeds too many concepts within single statements. The authors should consider simplifying sentence structure, improving paragraph transitions, and ensuring consistent terminology throughout. Professional English language editing would significantly enhance the manuscript's clarity and accessibility to an international readership.
Answer 6: Thank you for your valuable feedback. We have carefully reviewed the entire text with a focus on simplifying complex sentence structures, enhancing paragraph transitions, and ensuring consistent and appropriate use of technical terminology. Grammatical issues—including subject-verb agreement, article usage, and awkward phrasing—have been systematically addressed. The revised version has undergone thorough language editing by a native English-speaking academic with experience in scientific writing to further improve the overall readability and ensure the manuscript meets the standards of an international readership. We believe these revisions have substantially improved the clarity, flow, and accessibility of the study.

Round 2
Reviewer 1 Report
Comments and Suggestions for Authors
All requested revisions have been adequately addressed by the authors. I find the revised manuscript suitable for publication and recommend it for acceptance.
Reviewer 3 Report
Comments and Suggestions for Authors
I would like to sincerely congratulate the authors for their hard work in writing and revising the manuscript. Their dedication and perseverance have truly paid off, and I wish them continued success in their future endeavors.